# Subjective walkability perceived by children and adolescents living in urban environments: A study protocol for participatory methods and scale development in the WALKI-MUC project

**Daniel Alexander Scheller** *, **Joachim Bachner**

TUM School of Medicine and Health, Associate Professorship of Didactics in Sport and Health, Technical University of Munich, Munich, Germany

* daniel.scheller@tum.de

## Abstract

**Data Availability Statement:** No datasets were generated or analysed during the current study.

### Background

Youth's perceptions of a city or neighbourhood's walkability are important for determining the physical activity (PA) friendliness of their environment. Traditional objective measures of walkability fail to incorporate children and youth's (CY) subjective perceptions of places that they perceive as supportive for play and exercise. Internationally, the most promising subjective measure is the Neighborhood Environment Walkability Scale for Youth (NEWS-Y) questionnaire. Yet, the NEWS-Y is not available for German-speaking adolescents. In the WALKI-MUC project, a combination of participatory research methods is used to identify CY's perceptions of PA-friendly places in Munich, Germany. Based on the findings, a German version of the NEWS-Y (NEWS-Y-G) for subjective walkability measurement is developed.

### Methods and design

CY aged six to 17 years from neighbourhoods with different objectively-measured walkability, take part in photovoice, walking interviews and mapping to gather their perceptions of PA-friendly places. The participatory study begins with an introductory workshop and concludes with a follow-up workshop, where characteristics of PA-friendly places are discussed in focus groups. In between these workshops, participants complete a photo mission with a walking interview, allowing for individual significance of PA-related places to be shared in a one-on-one setting with the researcher. The findings are used to adapt and translate the NEWS-Y for the German context. The newly developed NEWS-Y-G is then used to measure the subjective walkability perceived by a representative sample of adolescents in Munich.

Deidentified research data will be made publicly available when the study is completed and published.

**Funding:** The WALKI-MUC project is funded by the Bayerisches Staatsministerium für Gesundheit, Pflege und Prävention via the Bayerisches Landesamt für Gesundheit und Lebensmittelsicherheit (grant number: K1-2497-GLB-21-V3-D23011/2021). The funders did not and will not have a role in study design, data collection and analysis, decision to publish, or preparation of the manuscript.

**Competing interests:** The authors have declared that no competing interests exist.

## Discussion

The WALKI-MUC project introduces a participatory methodology for researchers and urban planners to assess subjective walkability with CY. The combination of qualitative and quantitative walkability measurements is described in this study protocol. Findings on PA-friendly places contribute to environmental psychology and the development of the NEWS-Y-G adds a German-language instrument for subjective walkability measurement.

## Background

Physical activity (PA) is a vital behaviour for children and youth (CY) to grow up healthy [1]. In order to foster PA among the young, national and international recommendations regarding the amount of PA and how to promote PA have been developed and continuously refined [2,3]. Yet, 80% of CY worldwide do not meet the recommended time of at least 60 minutes of moderate-to-vigorous PA per day and more than 70% of 3 to 17 year olds were considered inactive in Germany [4,5]. One of the challenges to promote PA is to address the multiple levels of influence on behaviour embedded in the living environment of CY. In socio-ecological models, an individual's living environment is described by the policy, physical and social environment, which should all be considered by interventions that aim to change behaviour in sustainable manner [6]. Regarding the influence of the policy environment, school-based policies and policies to promote walking and cycling seem promising. Evidence on the effect of policies targeting the built environment and active travel, however, is ambiguous [7]. In terms of the physical environment, especially in cities, green and open recreational spaces disappear more and more, as do the opportunities to engage in PA and social interaction in these areas [8]. In line with this, CY show an increasing distance to natural environments [9]. Changes in the social environment might reinforce this trend with 70% of the world's population expected to live in urban areas by 2050 [10]. In Europe, this estimate is even higher with 80% of citizens expected to live in cities by 2030 [11]. Therefore, urban planning that facilitates PA is crucial for the development of CY, which is why research and politics have started to focus on neighbourhood environments with sufficient opportunities for PA [12].

### The young, their physical activity level and perceived environment

Physical activity is commonly described as any bodily movement carried out by skeletal muscles that elevates energy expenditure beyond resting levels [13]. In the context of the built environment, PA primarily pertains to the extent of bodily movement and energy expended during walking and other forms of active travel within a neighbourhood. This comprises, as examples, purposeful walking to work or recreational walking in the park, which in itself is beneficial for health [14]. On top of that, walking is a healthy means to reach public places, exercise or leisure destinations such as playgrounds, where CY can engage in all forms of PA, including those with a higher intensity level than walking. When studying PA behaviour in relation to the built environment, it is hardly possible to transfer the results of studies with adult samples to children. CY are less autonomous in their decision-making and their walking mobility is likely limited to neighbourhood boundaries, which is why their closer built environment might have an even greater effect on their PA behaviour compared to adults.

Some reviews highlight the positive effects of the built environment on youth's PA [15]. Other meta-analyses, however, only find small to moderate effects with different magnitudes

for children and adolescents [16]. This disparity can be well illustrated by the example of parks and playgrounds–places that are commonly considered as PA-supportive opportunities. According to reviews that focussed on the relationship between the living environment and the PA of CY, the presence, size and distance of parks and playgrounds affected children's PA level only in half of the studies [17,18]. For adolescents, there was hardly any association between green spaces and PA [18]. Still, when the association between CY's PA and their environment is examined, it is these types of places that are usually considered. A previous study in Germany examined the association of environmental variables, especially regarding availability of recreational facilities, i.e. playgrounds, green spaces and parks, with accelerometer-measured PA to create a moveability index that should represent urban opportunities for PA for children [19]. However, before valid evaluations on the PA-friendliness of neighbourhoods or even entire cities can be made, it is essential to know which places CY themselves actually consider as PA-friendly.

Perception of the environment depends on age and previous experience, personality and culture-specific as well as local environmental characteristics [20]. Local conditions of the natural and built environment and in particular their perception, i.e. the awareness of their existence, must be taken into account as they have a major impact on behaviour by providing so-called affordances. These are incentives that trigger people to behave in a specific way [21]. It is against this background that visual affordances in the environment can influence PA by creating opportunities and, from a negative point of view, barriers regarding movement, play or exercise. Another German index therefore incorporated qualitatively-assessed characteristics that determine the activity friendliness of urban neighbourhoods [22]. The authors identified supportive and limiting factors of activity friendliness from a subjective point of view by conducting focus groups with children in one area of Berlin, to later implement the collected factors in Geographic Information System (GIS). Apart from measuring PA-friendliness per se, it is important to understand why certain places are visited repeatedly for PA behaviour. As shown above with the example of green spaces that did not have the initially assumed relevance for CY's PA [17,18], the public's perception or that of researchers and practitioners regarding sites that might repeatedly attract CY and provide affordances for PA does not always correspond to the subjective perception of CY. From an environmental psychology perspective, place attachment is a concept that helps to understand the emotional bond between a person and a place. Theoretical frameworks like the Tripartite Model by Scannell and Gifford propose that person, process and place are the three main dimensions that affect place attachment [23]. Models like this should be taken into consideration when studying the environment of CY with the goal of promoting their PA in that same environment. A current study of Koohsari et al. showed that place attachment dimensions were significantly positively associated with neighbourhood-specific PA of adults in terms of time spent on walking for transport and recreation [24]. The associations between place attachment and walking for transport were mediated by the ways people perceived the built features in their neighbourhood. This has not yet been investigated for CY.

Regardless of whether CY's subjective perception is taken into account or objective approaches are used, there is generally little standardisation in what is being measured when trying to assess the PA friendliness of a city or neighbourhood, which also contributes to inconsistencies in previous literature regarding associations between environmental attributes and youth PA [18]. Koohsari et al. used measures of perceived neighbourhood walkability to operationalise features of the built environment [24].

## The concept of walkability and its measurement

In scientific research, characteristics of the built and natural environment that are related to individual PA are often summarized under the term "walkability" [25]. In the context of public

health, walkability is used beyond its literal meaning, i.e. to be able to walk in a certain area, and became increasingly important as a proxy for the activity friendliness of a neighbourhood or city [26]. Walkability can be understood in both a narrow and broad sense [27]. The broad understanding that is frequently used for CY goes beyond active transportation and walking and recognises the built and natural environment as spaces for people to linger, exercise and play. This perspective acknowledges that walkability is not solely about transport-related activities but encompasses the overall experience and functionality of the environment for PA with regard to aspects such as perceived safety and access to recreational facilities.

In terms of the assessment of walkability, both objective and subjective approaches can be used. Methods for walkability assessment can be divided into four groups: 1) GIS tools (objective); 2) street audit tools (objective/subjective); 3) methods based on surveys, interviews and questionnaires (subjective); and 4) mixed methods, which are a combination of the previously mentioned data sources [28]. Due to the high costs of conducting direct field research on subjective walkability (group 2 and 3), with audits being the most expensive and time-consuming, data from so-called secondary sources became the preferred choice. Main reason for this is that research suggests that street audit tools can be successfully replaced by using GIS tools based on secondary sources that are free of charge and unobtrusive, e.g. data collected by Google Earth [29] or Google Street View [30].

Objective Walkability measured by using GIS tools is usually calculated by the sum of z-scores of the parameters urban density, land use mix, street connectivity and distance to facilities. Since these parameters often highly correlate, a so-called walkability index has been developed to represent the parameters in one variable [31]. This index has been shown to positively correlate with walking and cycling for transportation in adults and also with their overall PA levels [32–34]. The walkability index and other objectively-derived walkability indicators like the publicly available Walk Score™ are frequently used in studies to find connections between health, walkability and other environmental parameters. Besides their free availability and resource-saving advantages, objective measures with GIS data imply substantial limitations in studying CY's environments as they often fail to differentiate between specific target groups and their respective perceptions. This also leads to ambiguity in empirical findings. For instance, Schicketanz et al. suggest that low intersection density had a positive effect on children's PA [22], which is in direct contrast to evidence generated by studies using the walkability index [31]. This shows that, although these objective methods initially appear to be an economic alternative, they cannot replace the strengths of subjective approaches. Instead, both approaches should be used in combined manner in order to include subjective perceptions and to reduce the impact of measurement mode [35]. After conducting a review about walkability measures in paediatric populations, Ubiali et al. stated that written surveys and interviews (group 3) are currently the most frequently used approaches in subjective walkability research. In the spectrum of subjective survey methods, both qualitative and quantitative methods are used.

Quantitative instruments to assess subjective walkability pose a difficult task for researchers, as they have to operationalise environmental characteristics, e.g. land use mix and street connectivity, based on individual perceptions. A vast majority of studies included in a review about walkability and CY's health evaluated one or more of the following nine components for measuring walkability: diversity of land use mix, neighbourhood recreation facilities, residential density, accessibility of land use mix, street connectivity, walking/cycling facilities, neighbourhood aesthetics, pedestrian and road traffic safety and crime safety [36]. Brownson et al. described a great number of available instruments to measure perceived walkability based on these variables ranging in length from seven to 68 items [20]. A very well-known self-report questionnaire that incorporates all of these components is the Neighborhood Environment

Walkability Scale (NEWS) [37]. Cerin et al. developed an abbreviated version (NEWS-A), which maintained its structure but reduced the number of items from 68 to 54 [38]. Relevant environmental characteristics for youth were then added to the NEWS-A to develop the NEWS-Youth (NEWS-Y) with 67 items [39]. The NEWS-Y has already been translated into various languages and adaptions were made to enable its application in countries participating in the IPEN study (NEWS-Y-IPEN) [40]. In 2013, Reimers et al. conducted a systematic literature review to identify self-report and parental questionnaires to measure the neighbourhood environment walkability for children and adolescents and found that most of the original scales were English and embedded in more comprehensive questionnaire instruments [41]. They developed a strongly shortened German scale with eight items based on the NEWS-Y [41,42]. However, besides this shortened NEWS version for youth and a regular version for adults [43], there is no comprehensive NEWS instrument that can be used for youth in the German context, which has already been criticised in 2014 by Rottmann and Mielck [44]. According to both Ubiali's and Yang's reviews in 2021, the variance in variables measured and tools used for walkability research in CY remains an issue [35,36]. Out of all available instruments, the NEWS-Y is the most popular one, and thus offers the greatest opportunities for international comparisons. To accommodate the use of the NEWS-Y in Germany, the tool including all its items should be translated and adjustments to environmental characteristics that are of specific relevance for CY in Germany should be considered.

Qualitative approaches for subjective walkability assessment are essential to adjust quantitative measurement tools according to local characteristics. Measurement instruments that attempt to quantify subjective walkability might neglect important qualitatively assessable aspects, e.g. perceived characteristics of the environment like safety and aesthetics, that could significantly affect young people's willingness to walk and exercise [36]. Tools should thus not only be age-specific in terms of language but should also address as many aspects as possible that may be relevant for the respective age group. He et al. therefore have added new items to the NEWS-Y in order to capture Hong Kong-specific environmental attributes and adapt it for Chinese children (NEWS-CC) [45]. The items were generated in a previous study using Nominal Group Technique with 34 children aged 10 to 11 years from four types of Hong Kong neighbourhoods varying in socioeconomic status (SES) and objective walkability (estimated through household density) [46]. The children named 30 aspects that they perceived as important for their willingness to engage in PA, which were then further categorised. Nine of these aspects were classified as relevant to children's PA and integrated as new items into the NEWS-CC, mainly to the aesthetics subscale [45]. For adults, interview techniques have been increasingly used to explore how people's experiences and perceptions of their community environments affect engagement in PA and their findings have also been summarised in reviews [47]. However, for the young, approaches from the qualitative spectrum were extremely underrepresented in Ubiali's review with only one study (0.5% of the included studies) that used map-drawing exercises [35]. Mapping approaches working with drawn pictures have shown to be promising in examining children's perceptions of places to play and be physically active [48]. However, quality of the findings of this approach depends on participants' drawing abilities. More current image-based approaches use photographs to study PA-related environmental research questions with photovoice being one of the prominent methods [25]. In this photo elicitation method, participants are equipped with cameras and "missions" to take photographs related to the research objectives, in order to later discuss them in focus groups. Both images and texts describing these images are generated as materials for scientific analysis, e.g. to identify opportunities and barriers for PA among residents [49].

## Participatory walkability research–perspectives and refinements

Photovoice has been successfully used in participatory health studies and has raised great enthusiasm for future research [50,51]. Participants take pictures during their photo missions and later discuss them together with researchers. Individuals who are often underrepresented or only "objects" of studies, such as CY, are empowered by having the opportunity to play an active role in the research process [52]. Contrarily to purely language-based methods, the focus on the visual enables research at eye level by eliminating the role of differences in vocabulary and knowledge of participants and researchers. Unlike interviews or focus groups, where an immediate response is expected, research participants are given time to think about their photos, and thus are encouraged to actively conceptualize and contemplate. This reflection period, or, the time for the photo mission, varies with the study design but usually does not exceed one week [53,54]. Photovoice has become a popular approach in participatory research with CY [52,55,56]. A study in South Africa combined photovoice with so-called community mapping to include another group exercise next to the photo discussion in the participatory process [53]. In community mapping, participants jointly draw a map of pictures or themes related to their neighbourhood. Depending on research topic and chosen design, this can include geographic as well as more abstract characteristics of a community [57]. This combination helped to visualise children's representations and perceptions of natural spaces first individually and then in the group [53].

A shortcoming of photovoice studies is the time between data collection, i.e. the moment a person has taken a photo, and the time it is discussed in the focus group workshop. Longer periods may lead to a recall bias and certain details, emotions and feelings attached to the photograph might be forgotten by some participants. In these cases, the actual advantage of high ecological validity data in form of "live" recorded environmental perceptions over nominal group techniques or written surveys in classroom atmosphere might get lost. This potential limitation can be avoided by having an interview partner who accompanies the participant and is thus able to document the rationales behind the photos in the moment of exposure. These so-called walking interviews involve a researcher walking with one or more participants while conducting and recording an interview. This produces rich narratives of places both in terms of their quantity and spatial specificity to the study area [58]. Walking interviews as a separately used research method have already been frequently used in ethnography, anthropology, and geography. At the same time, there have been suggestions for their enhancement through mapping and additional techniques [59].

Loebach and Gilliland have used photo elicitation combined with child-led tours to explore the use of neighbourhood environment in London, Canada, by having student pairs accompanied by two adult observers who were able to document the rationales behind the photos while the students were taking them [60]. On self-selected walks around their neighbourhood, 16 children highlighted their "favourite" and "least favourite" community places. The two authors independently reviewed all of the children's narratives and consecutively selected photographs to draw up a list of emerging themes and patterns. One week after the child-led tours, the research team returned to the school and presented the photographs together with the identified themes in order to have them verified or adapted by the students in a group discussion. According to the authors, a "potential drawback to the group discussion is that children were not necessarily given the opportunity to discuss the personal significance of their individual photographs". In addition, the participants might have been strongly influenced by third parties in both activities. Not every student might really have selected the walking routes completely by him or herself while being observed by one peer friend and two adult researchers. In the group discussion, the students might have been reluctant in correcting the themes

that had been identified by the adult researchers by means of an ad-hoc analysis previous to the focus group discussion. A collaborative mapping exercise, as it was done in the study by Adams et al., could have helped the themes to emerge from the group [53]. These considerations suggest that combining participatory methods can have tremendous benefits for research, given that they are combined in a way that leverages their respective strengths and compensates for their weaknesses.

Facing population growth and residential densification, European cities now draw their attention to accessible public spaces that are significant for CY's play and other PA. These spaces are necessary for a PA-friendly environment regardless of what kind of PA they promote or to what extent. These PA-friendly places, sometimes referred to as "third places" in a neighbourhood next to home and school, have an impact on PA and can be divided into destinations (e.g. parks, shops), threshold spaces (semi-public areas, e.g. grass verges), and transitory spaces (routes that link children's daily destinations, such as footpaths) [61]. Appropriate methodological approaches are needed that help to identify which third places and opportunities for PA exist in walking distance. Thereupon, the characteristics of these places should be included in quantitative instruments, which are designed to measure perceived walkability or PA-friendliness in the same area. The NEWS-Y has shown its suitability in the US, where many cities were planned top-down to provide a good walkability but it might be adapted to the local characteristics of organically grown European cities with different urban structures. He et al. have shown that the NEWS-Y provides a solid basis for the subjective assessment of walkability when tailored to the region with the help of CY included in qualitative studies [45]. However, to conduct such research on a city level, several groups of different age and from different neighbourhoods and sociodemographic backgrounds have to be included. Especially for CY from lower socioeconomic backgrounds, the closer living environment may have a more important impact, as studies have shown that they have less access to structured exercise and sport, but spend more time in active play around home [62,63]. Furthermore, the focus must remain on the young individual who has to be able to express his or her opinion freely in the data collection process and analysis. A combination of photovoice, walking interviews and mapping exercises applied with CY of different age and socioeconomic backgrounds could enable participants to share their thoughts about "walkable places" in their immediate environment. In Munich, Germany, where a population growth of 14% in the period between 2022 and 2040 has been forecasted [64], the research project "WALKI-MUC" applies this participatory approach for the evaluation of walkability based on the perception of CY living in selected neighbourhoods. The walkability index Munich already allows for differentiating between city areas of diverse objectively-measured walkability [65,66]. These districts or neighbourhoods should be further examined by subjective measurements of perceived walkability from the perspective of young people. Consequentially, the aims of the WALKI-MUC project are:

1. To identify characteristics of walkable PA-friendly places in Munich neighbourhoods of different objectively-measured walkability with a sample of CY between six and 17 years of age by using a qualitative method combination of photovoice, walking interviews and mapping exercises.

2. To translate the NEWS-Y for German-speaking adolescents aged eleven to 17 years (named NEWS-Y-G) and to adapt the instrument to local characteristics based on the findings on walkable PA-friendly places.

3. To pilot the NEWS-Y-G in a validation study and to conduct a cross-sectional evaluation of perceived walkability with a sample of adolescents living in neighbourhoods of different objectively-measured walkability.

This study protocol describes the qualitative and quantitative methods to address these research aims.

## Methods/Design

### Study aim and design

The WALKI-MUC project takes place in Munich, Germany, and includes two study phases that build on each other (Fig 1). Study phase one entails the combinatory use of participatory research methods with photovoice, walking interviews and mapping exercises across at least three different neighbourhoods in central urban and lateral urban locations with children aged six to 17 years. Main aim of the first study phase is the inductive identification of places that CY perceive as PA-friendly by use of a combination of three qualitative methods that complement each other with visual (photovoice), verbal (walking interviews) and textual/geographical (mapping exercises) content. By definition, the places should be 1) reachable on foot from home by CY and 2) supportive for physical activity, i.e. places perceived by CY to be suitable for play, exercise and PA. In the following, these places are referred to as "walkable places". Based on the generated data, the following research questions are examined:

- What are CY's perceptions regarding walkable places in their respective neighbourhood?

- Which are the main characteristics of walkable places that have potential to establish place attachment in CY?

- Are the identified characteristics of walkable places taken into account in the calculation of objective walkability, e.g. in the walkability index?

| | Phase 1: Qualitative study | | Phase 2: Quantitative study |
|---|---|---|---|
| **AIM** | **Assessment of place attachment to physical activity-friendly places in neighbourhoods of Munich with different objectively-measured walkability** | **Adaption of the Neighborhood Environment Walkability Scale for Youth (NEWS-Y) for the German context** | **Evaluation of subjective walkability with NEWS-Y-G in neighbourhoods of Munich with different objectively-measured walkability** |
| **ACTIVITIES** | • Photovoice<br>• Walking interviews<br>• Mapping exercises<br><br>Participants aged 6 to 17 years (n=90) | 1) Back-translation technique<br>**2) Adaption of NEWS-Y**<br>3) Validation study in test-retest-design<br>Participants aged 11 to 17 years (n=200) | Online survey with representative sample of adolescents in Munich<br><br><br>Participants aged 11 to 17 years (n=600) |
| **OUTCOMES** | • **Perceived characteristics of physical activity-friendly places**<br>• Criteria for place attachment (place, person or process)<br>• GPS data | • Validated Neighborhood Environment Walkability Scale for Youth German (NEWS-Y-G) | • Measurements of subjective walkability<br>• Associations with physical activity, socioeconomic status and objectively-measured walkability in Munich |

**Fig 1. The WALKI-MUC project.** Based on the qualitative results from the first study phase, the NEWS-Y-G for quantitative measurement of subjectively-perceived walkability is developed, validated and applied in the second study phase.

Study phase two starts with the development of the German Neighborhood Environment Walkability Scale for Youth based on the results of the first study phase. Therefore, the English NEWS-Y is translated into German (NEWS-Y-G) and, as far as necessary, adapted according to the characteristics of walkable places identified in study phase one. After a successful validation, the NEWS-Y-G is used for a cross-sectional survey with a representative sample (according to age and SES) of adolescents aged eleven to 17 years living in Munich. The research questions of the second study phase are as follows:

- Does the NEWS-Y need to be adapted to the German context according to the qualitative findings of study phase one?

- Does the NEWS-Y-G exhibit satisfactory reliability and validity?

- How is the subjective walkability perceived by adolescents in selected neighbourhoods of Munich when measured with the NEWS-Y-G?

- What are the associations of subjective walkability with self-reported physical activity, SES and the objective walkability index?

As described, perceptions of the neighbourhood are influenced by individual factors such as age, sex, beliefs or memories [20]. In addition to this individual environment, socio-ecological models describe factors in social, physical (built and natural) and policy environments, all of which determine PA behaviour [6]. With regard to this environmental framework (Fig 2), our aim is to first understand CY's perceptions on walkable PA-friendly places and then measure how CY subjectively perceive neighbourhood walkability.

The study protocol has been approved by the ethics committee of the Technical University of Munich (reference number 77/22 S).

## Study area

The neighbourhoods of Munich are organically grown areas that are officially divided into 25 administrative districts. Below this district level, there are 475 subdistricts. Based on the objective walkability indices that are available for these subdistricts, a mean index can be calculated for the 25 administrative districts. All Munich districts, or neighbourhoods, are consequently classified in five categories ranging from "very low" to "very high" objectively-measured walkability. This aggregation of the walkability index on district level can only offer a tendency about the objective walkability in a district, because the small-scale indices of the subdistricts can vary significantly within a district. The neighbourhoods that form part of the study are selected in a way that allows to cover the whole spectrum from "very low" to "very high" objective walkability.

Moreover, the Social Department of the Munich city government offers socioeconomic data that can be considered during the neighbourhood selection process. Their monitoring system incorporates the "social challenges" indicator, which ranges between one (very low) and five (very high) and is again based on diverse indicators such as district social work, unemployment rate, social benefits and basic security in the neighbourhood [67].

## Phase 1: Qualitative method combination of photovoice, walking interviews and mapping exercises

**Study participants, sampling and recruitment.**   Approximately 30 participants aged six to 17 years from at least three neighbourhoods with different objective walkability are included. This results in a total of 90 participants in the first study phase. Participants are recruited via the so-called "Bildungslokale", which are local institutions in a neighbourhood

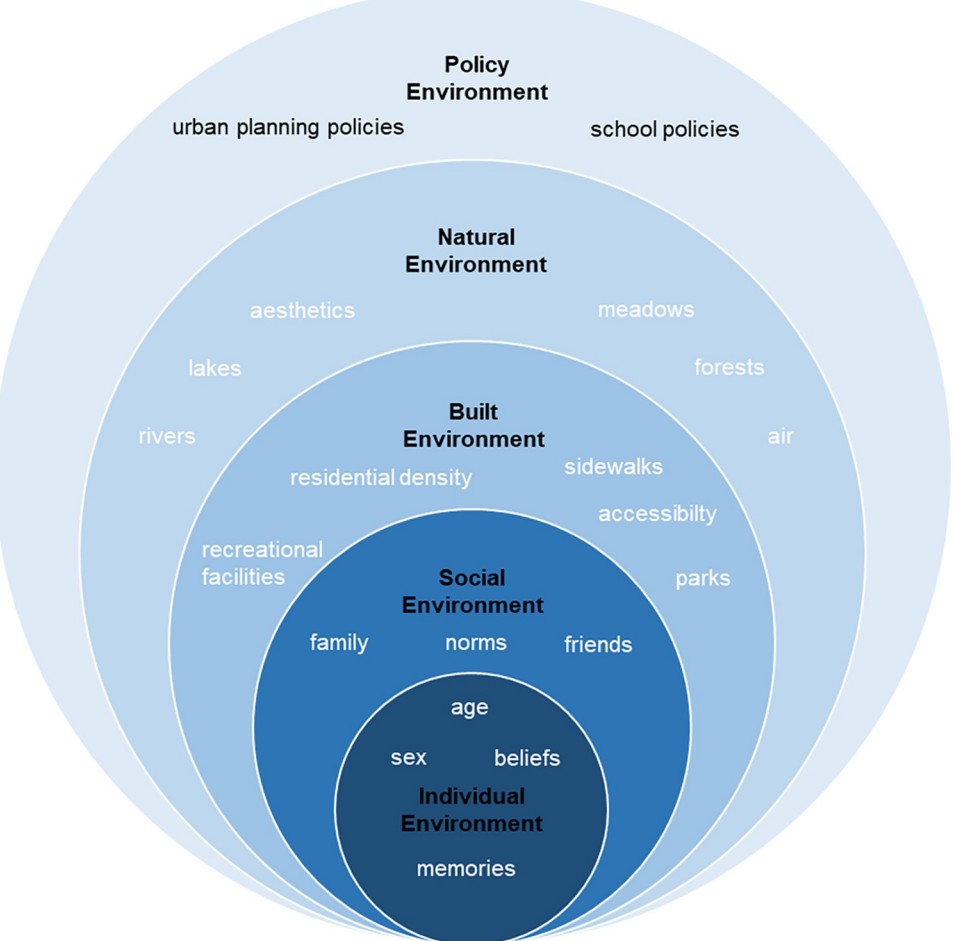

**Fig 2. Conceptual framework of the study based on the socio-ecological model for physical activity [6].** This modified socio-ecological model introduces subjectively-perceived environmental factors on different levels that can both hinder or facilitate an individual's physical activity in the neighbourhood.

that offer free educational counselling for all citizens of a neighbourhood. There are eight of these facilities in Munich, primarily in neighbourhoods with a higher need for educational support. This aspect also makes the "Bildungslokale" popular contact points for families with a migration background. The recruitment strategy ensures the inclusion of CY from different ethnic and socioeconomic backgrounds in the study. The "Bildungslokal" managers are educational actors who are closely networked with all types of schools and other children- and youth-related facilities of a neighbourhood. These managers are contacted and asked, if other educational actors (e.g. teachers or caregivers of afternoon care) with potential participants of different age would be willing to take part. Ideally, researchers and educational actors who are responsible for leading the participating group, schedule a meeting prior to data collection to facilitate the upcoming data collection. At this meeting, researchers get to know the facilities and explain the procedure in detail to the educational actor who then forwards project information and consent forms to the potential participants. For participation, one parent/legal guardian and the participant have to sign a declaration of consent. Privacy policies concerning audio-recorded material and photographs are handed out and have to be consented to by signature. Further, all participants are fully informed about their rights to withdraw from the

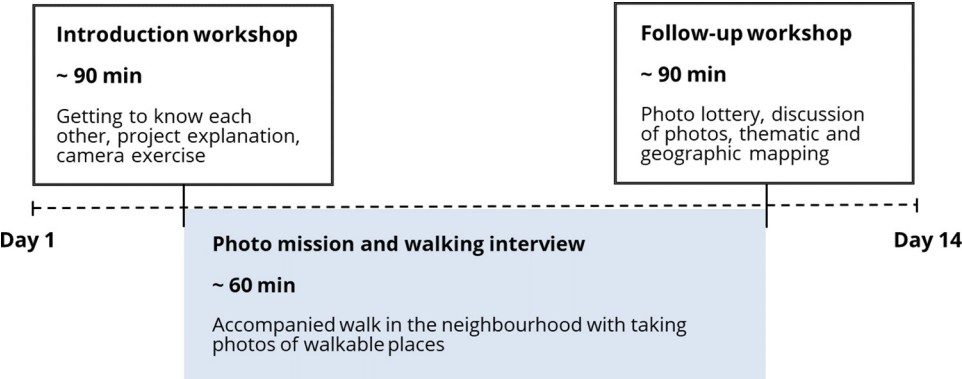

**Fig 3. Procedure in study phase 1.**

project at any time without any consequences. Written consent is obtained and documented from all participants for their inclusion in the study.

**Data collection.** Data is collected using three participatory methods, namely photovoice, walking interviews and mapping exercises. In each neighbourhood, participants are divided into age groups of six to ten and eleven to 17 years with a maximum of 15 participants per group. A group is composed of CY of diverse age, gender and cultural background. The first study phase consists of two workshops with the elements of training, taking photographs, reflective discussing and sharing, starting with an introduction workshop and ending with a follow-up workshop. In between these workshops, a photo mission with walking interview takes place (Fig 3). Workshops are led by at least two researchers, whereas the walking interviews are conducted by one researcher in a one-on-one situation. The data collection period takes several months due to school holidays and the rather time-intensive attendance of participants. Participation in the two workshops and the walking interview take about 240 minutes in total within a time span that can last from a minimum of seven days to a maximum of 14 days, depending on the availability of the participants. Whereas previous photovoice studies have often included multiple sessions, in WALKI-MUC there is only one introduction workshop and one follow-up workshop. This is mainly a practical decision to accommodate the participants' schedules and to ensure their continued engagement. This practical decision is supported by the time-intensive nature of the walking interviews. However, it is precisely these interviews that outweigh the smaller number of photovoice sessions as they offer an in-depth understanding of CY's subjective perception in a one-on-one setting. By combining photovoice, walking interviews, and mapping exercises, detailed insights can be gained within an economic timeframe that is feasible for this specific sample.

**1. Introduction workshop to photovoice (estimated duration: 90 minutes).** In the first workshop, researchers and participants meet and discuss the goal of photovoice. Photovoice is a participatory method in which people use cameras to document their perceived reality regarding a specific topic [52]. In this study, photovoice is used to understand CY's perceptions of walkable places and to identify the characteristics of these places. The workshop starts with an age-appropriate icebreaker game to establish a personal connection between researchers and participants and to ease the atmosphere. Generally, it is considered important to find a balance between research activities, breaks and games to get to know each other while keeping the focus on the research topic. Besides establishing a personal connection, the main aim is to explain the long-term purpose of the project, which is to evaluate the PA-friendliness of the participants' city from their own perspective. In an opening presentation, current

developments are shown (e.g., progressing urbanisation and densification) with particular focus on the respective neighbourhood. It is emphasised that (young) residents need to help to ensure that spaces for PA are preserved. Thereupon, participants are asked to think about their favourite and least favourite places in their neighbourhood with regard to play, exercise and being physically active. Typically for photovoice projects, they are encouraged to develop their own ideas about the characteristics a place needs to have in order to be considered as PA-supportive. These ideas contribute to a collective basis for the upcoming research [68]. It is made clear that the perceived characteristics of a walkable place can vary between individuals and that participants can choose any place they consider as PA-friendly. They are informed that taking pictures of these places in the next step of the project represents their so-called photo mission. Ethical considerations of photovoice (e.g., verbal consent to photograph other persons) are introduced before a practical photography training with Polaroid cameras follows. In small groups, the participants take test photos in the surrounding area of the respective facility, whereby the motive can be chosen freely and should be incentivising (e.g., photo of a friend), as this training should contribute to motivation and knowledge about the cameras. The produced Polaroids of the training can be taken home as incentive. In the last part of the first workshop, an appointment is made with each participant to pick him or her up at home or at a desired location near home for the photo mission and walking interview. For the younger participants (six to ten years), the appointment is made directly with the parents. Participants and their parents have to be introduced to the researcher who accompanies the young interviewees during their photo missions and contact information is exchanged. This happens either directly after the first workshop or at the beginning of the second meeting, the photo mission.

**2. Photo mission and walking interview (estimated duration: 60 minutes).** The interviewer has a short checklist containing the most important steps for preparing, conducting and post-processing the walking interview. The questions and content of the interview is guided by a semi-structured interview protocol based on the "SHOWeD" questions of the photovoice study by Wang and Burris [52]:

What do you **S**ee here?          → Describe the place

What is really **H**appening?          → Describe activities at the place

How does this relate to **O**ur lives?          → Describe frequency of visits and people at the place

**W**hy does problem or strength exist?          → Describe weaknesses and strengths of the place

What could be **D**ifferent?          → Describe potential improvements of the place

The participant is picked up at home or the agreed pick-up location by an interviewer who is preferably of the same gender. Immediately after the greeting, it is recommended that the participant has a short period of time to get used to the interviewer, especially if the interviewer was not present at the first workshop. During this period of about five minutes, no audio recording should be done. This familiarisation phase can take place in the presence of the parents or at the beginning of the walk through the neighbourhood. It is advisable for the interviewer to give clear instructions to the participant on how the interview is conducted and to clarify the respective roles. The interviewer has to keep the conversation going while asking the questions according to the interview protocol. The participant is responsible for the places they visit and the route they take to get there. After the objectives of the photo mission are clarified once again, the audio recording can be started with the note that sections of the interview can be deleted if the participant says something that he/she would not like to be further processed afterwards. For the audio recording of the interview, the participant carries a dictaphone around the neck. Further, the interviewer tracks the route by using GPS. Each

participant is asked to show the interviewer at least two places in his/her neighbourhood and to take pictures of them with the Polaroid camera. If a participant has not thought of a location before the start of the photo mission, guidance is given again by the interviewer to choose places of the following three categories for participants:

1. their favourite walkable place where they like to play, exercise and be physically active

2. their least favourite walkable place that they think is designed for them to play, exercise and be physically active, but which they usually do not like to use; or
where they would like to play, exercise and be physically active, but are not allowed to

3. relevant walkable places for them that are not their favourite or least favourite places

It is up to the participant to decide which place to visit first. The questions about the place should already be asked while walking towards the place and then become more concrete at the place itself. The interviewer uses a smartphone to photograph the places and to add geo-tags for having the respective location available. The interviewer should ensure that the smart-phone photos replicate the participant's Polaroids, e.g. in terms of the position from which the Polaroid was taken, in order to capture the exact perspective of the participant in this moment. In this way, the participant can take the Polaroid pictures home and the research team already has a digital collection of the images including a spatial understanding of the location of photographed places in the neighbourhood. The interview ends with bringing the participant back home or to the starting point. The Polaroids are handed over to the participant, with the reminder to bring them to the upcoming second workshop, where the photos are discussed. The interviewer is required to send the digital pictures on the smartphone to the lead researcher together with short narratives from the walking interview according to the five SHOWeD questions. The research team prepares a presentation and digital map that shows the covered distance of the study group. This presentation serves as a starting point in the upcoming follow-up workshop with the whole study group.

**3. Follow-up workshop with photo lottery and mapping exercises (estimated duration: 90 minutes).** To prepare the follow-up workshop, smartphone pictures are classified according to what they were referred to in the interview by the participants. Categorization is illustrated by use of a traffic light system: 1) favourite or good walkable places are marked in green, 2) least favourite or bad walkable places are marked in red and 3) other walkable places that can be either good or bad are marked in yellow. As initiating workshop activity, the research team presents slides that each show a compilation of all green, red and yellow places of the group, respectively. Additionally, a map with the recorded routes that participants took gives an overview of what they have achieved. Following the presentation, a so-called "photo lottery" is conducted. For this lottery, all Polaroid pictures are collected and labelled by writing a uniquely identifiable code on the back of each image. Then, Polaroids are put in a container, e.g. a box or bag, and each participant blindly selects two Polaroids from the container. If a participant draws his/her own Polaroid, he/she must put it back in the container and draw again. One after the other, participants name the code on the back of the drawn images and describe them following the SHOWeD questions [52]. Researchers make notes on the discussion of each image. After that, the participant tries to interpret the places as green, red or yellow places, which is subsequently resolved by the participant who took the picture, i.e. the photographer. The photographer is further given the opportunity to tell the group why he or she has chosen this place. This self-invented exercise is intended to serve the purpose of making the analysis easier and more playful for the young participants who might be initially reluctant to present their own images. Besides, everyone is given the opportunity to discuss the personal significance of their individual photographs to the extent they want and peers can

add new information through the reflection of different perceptions on the same place. This exercise should be conducted in an appropriate discussion atmosphere, e.g. in a sitting circle, that further contributes to the next activities. In the following, two mapping exercises are conducted that contribute to the analysis of the pictures. Wang and Burris recommend the thematic mapping, or "codifying", as part of the common photovoice analysis [52]. The aim is to identify overarching themes that can be found across all pictures. Therefore, all pictures are laid out on a table in order to be visible for everyone. Sticky notes and pens are handed out together with the instruction to group the pictures according to the characteristics of the places. Participants group all Polaroids of a theme together and write down a name for each picture group on a sticky note that is placed next to the group. A picture can be assigned to several themes, i.e. picture groups, and thus participants are invited to re-arrange the pictures in as many groups as they want. This also means that several aspects (themes) can be considered on one given picture. All created picture groups including their theme name on the sticky notes are photographed by the researchers. The collection of themes is illustrated in a mind map of the sticky notes afterwards. Additionally, the spatial relationships of the photographed places are worked out. Therefore, a geographic mapping exercise is conducted with neighbourhood maps designed for children. Participants can locate their photographed places and draw their travelled routes onto these maps. To round it up, the workshop ends with a brief reflection on the project and evaluation of the generated outcomes.

**Pilot study in phase 1.**   The qualitative methodological approach is pilot tested before the start of the main study. The pilot study takes place in one neighbourhood with one group of children (aged six to ten years) and one group of adolescents older than ten years, each comprising around 15 participants. By implementing the mentioned strategies (such as maintaining small groups and utilising a photo lottery), an environment that is inclusive for all participants is established. After the pilot phase, the efficacy of these strategies are evaluated and adjusted, if necessary (e.g. separation of participants according to gender). The method combination of photovoice, walking interviews and thematic/geographic mapping exercise is examined for its feasibility. Walking interviews are transcribed verbatim and an inductive thematic analysis is conducted according to the principles of Braun and Clarke. Furthermore, CY's statements in response to the SHOWeD questions as well as the outcomes of participants' thematic mapping are evaluated. The feasibility of the applied methods is qualitatively analysed following a structured framework [69]. The dimensions of *process*, *resources*, *management* and *scientific* are evaluated to result in one of four possible feasibility outcomes: "(i) Stop–main study not feasible; (ii) Continue but modify protocol–feasible with modifications; (iii) Continue without modifications but monitor closely–feasible with close monitoring and (iv) Continue without modifications–feasible as is" [69]. Modifications to improve the main study are made accordingly. Researchers who conducted the walking interviews can share their experiences in an online survey and rate the feasibility of the walking interviews. Acceptability of the workshops from the perspective of participants and educational actors is assessed with questionnaires.

**Main study analysis of walkable places and their characteristics.**   Once the desired number of 90 participants from three different neighbourhoods is reached, all walking interviews are transcribed verbatim and analysed in MAXQDA Analytics Pro 2022. For the qualitative analysis of interviews, there are two options: 1) inductive thematic analysis after Braun and Clarke like in the pilot study or 2) deductive coding according to the Tripartite Framework of place attachment [23,70]. For the latter option, the research team developed a codebook based on the Tripartite Framework. The walking interviews of the pilot study are analysed by use of both options. If deductive coding with the Tripartite Framework is feasible and if its dimensions match the inductively identified themes of the thematic analysis after Braun and Clarke

[70], the deductive approach for qualitative analysis following the Framework Method [71] is preferred due to the large amount of data expected in the main study. The themes from the thematic mapping exercise are used to complement the data from the walking interviews. The findings on characteristics of walkable places are the foundation for the adaption of the NEWS-Y-G in the second study phase.

## Phase 2: Development of the NEWS-Y-G and cross-sectional evaluation of subjective walkability in Munich

**Study participants, sampling and recruitment.** In the second study phase, subjective walkability perceived by adolescents is measured by using the newly developed NEWS-Y-G. In contrast to study phase one, where data is collected in-situ, only adolescents aged eleven to 17 years are included, as children might not have the spatial understanding for answering abstract questions about their neighbourhood without seeing it "live". For validating the NEWS-Y-G, about 200 participants are included. In the subsequent evaluation of subjective walkability, the NEWS-Y-G is conducted with a sufficiently large sample size of about 200 participants per neighbourhood, which is in line with previous assessments with newly developed NEWS tools for adolescents [40]. Participants are recruited from the neighbourhoods that were included in the first study phase, adolescents from other neighbourhoods are added. It is ensured that participants stem from neighbourhoods that substantially vary in their objective walkability index. First, participants are recruited via the networks that have been built with the educational actors who participated in study phase one. Further participants are recruited with the help of the city's registration office. Random sampling is applied with regard to objectively-measured walkability, SES and age. Potential participants receive project information and consent forms via educational actors or directly by mail or e-mail. For participation, one parent/legal guardian and the participant have to sign a declaration of consent. Again, all participants are fully informed about their rights to withdraw from the project at any time without any consequences. Written consent is obtained and documented from all participants for their inclusion in the study.

**Data collection.** In each neighbourhood, access to the NEWS-Y-G questionnaire is provided to adolescents directly (sent by e-mail or with a pre-paid return envelope) or to educational actors for further dissemination. NEWS-Y-G surveys are self-administered in paper-pencil or online. Participants are asked to complete the questionnaire without assistance. Besides the NEWS-Y-G, data collection includes assessment of further aspects, such as demographic variables, PA and body mass index. The demographic variables include gender, age, SES and school type. SES is estimated with regard to the parents' current jobs by using the International Socioeconomic Index of Occupational Status (ISEI) based on the International Standard Classification of Occupation 2008 (ISCO-08) [72]. If both parents are currently employed, the job with the higher ISEI is considered (HISEI). PA is measured with the International Physical Activity Questionnaire (IPAQ) [73]. Average duration to answer the NEWS-Y-G is about 20 minutes. This estimate is verified in the pilot study.

**Adaption of the NEWS-Y for German-speaking adolescents.** The NEWS-Y-G is developed based on the original English version of the NEWS-Y [39]. Following an approach that is similar to the one used by He et al., the NEWS-Y is adapted by first translating the tool from English to German by using a back-translation technique [45]. By using Flaherty's 3-point scale [74], the English back translation is checked for semantic equivalence with the original English questionnaire to evaluate whether each item has the same meaning after translation. The NEWS-Y measures walkability with nine subscales: diversity of land use mix, neighbourhood recreation facilities, residential density, accessibility of land use mix, street connectivity,

walking/cycling facilities, neighbourhood aesthetics, pedestrian and road traffic safety and crime safety. Items of most subscales offer a four-point Likert scale except for residential density (five-point Likert scale) as well as land-use-mix and recreational facilities, which both represent checklists. In these checklists, it is asked for the perceived walking distance from home to different types of destinations that are relevant for adolescents. The five response options for each destination range from one to five minutes of walking distance to over 30 minutes. Items measuring proximity to destinations that are not deemed to be highly relevant to adolescents in the German context are omitted to reduce the length of the questionnaire. This approach was also applied in the development of the NEWS-IPEN [40]. Besides this, items are modified if they are considered inappropriate for the German context (e.g. YMCA as recreational facility). Additionally, new items can be added in the form of destinations identified as relevant for German adolescents in the first study phase.

**Validation of the NEWS-Y-G.**   After the translation of the original English NEWS-Y for the German context and its adaption according to findings of study phase one, the resulting NEWS-Y-G is validated with a sample of adolescents. Preferably, the questionnaire is filled in paper-pencil-based and on site (e.g. in education centre or schools). Participants are asked to advice the research team about the applicability of the questionnaire. Items that they cannot understand or seem inappropriate are discussed within focus groups. In these discussions, adolescents are given the chance to help re-word the items. The NEWS-Y-G validation follows the procedure of prior NEWS developments [39,40,45]. Factorial validity of the NEWS-Y-G is examined in a confirmatory factor analysis (CFA). All items are organised in an a priori model according to the structure of the original NEWS-Y with six subscales using a four-point Likert scale and three subscales using other response formats. The items of the three subscales without four-point Likert scale do not load on a latent factor, which is why they are not included in the CFA. Since the sample provides multi-level data (individual participants nested in neighbourhoods), intraclass correlation coefficients (ICCs) are calculated for each NEWS-Y-G item to estimate the proportion of variance that is explained by neighbourhood affiliation. An average ICC value larger than 0.10 indicates the need to conduct a multilevel CFA, otherwise a single-level CFA is be done [38]. The fit of the a priori model is tested with the comparative fit index (CFI), the nonnormed fit index (NNFI) and the goodness-of-fit index (GFI), which should be larger than 0.90. As additional fit indices, the standardized root mean squared residual (SRMR) should be less than 0.10 and the root mean square error of approximation (RMSEA) should be less than 0.06. If a priori model does not exhibit an appropriate fit, factor structure of the NEWS-Y-G is respecified according to the results of the initial CFA and in line with theoretically meaningful considerations. CFA is performed on the final NEWS-Y-G to re-examine its factorial validity. Internal consistency of subscales is assessed using Cronbach's alpha. Reliability of the NEWS-Y-G is assessed using a test-retest design, following the procedure of the original NEWS-Y, where in average there have been 27 days between two completions of the survey. The time between the two measurement points of the NEWS-Y-G is intended to be at least one week but should not exceed three weeks. Test-retest reliability of the NEWS-Y-G is determined by calculating one way single-measure ICCs of subscales and their included items. The interpretation of ICC values for reliability is based on Landis and Koch: 0.00–0.20 as slight, 0.21–0.40 as fair, 0.41–0.60 as moderate, 0.61–0.80 as substantial, and 0.81–1.00 as almost perfect reliability [75]. Items with an ICC value of 0.40 or less are excluded.

**Analysis of associations between subjective walkability measured with the NEWS-Y-G and physical activity, socioeconomic status and objectively-measured walkability in Munich.**   The subscale scores of the NEWS-Y-G are calculated following the NEWS-Y scoring guidelines [39]. Correlations are used to examine associations of the NEWS-Y-G subscale

scores with adolescents' self-reported PA levels and SES. Using subjective measures to assess perceptions of neighborhood features and correlating them with participants' self-reported PA levels may imply a same-source bias [76], since the way an individual perceives the environment may affect the estimate of how physically active he or she is in this environment. Strategies to prevent a potential same-source bias in the study are to 1) aggregate ratings on subjective walkability of multiple CY within the same neighbourhood [77], 2) control for contextual factors that may impact same-source bias, including participant demographics and SES 3) supplement the self-reported NEWS-Y-G data with data of official administrative sources, such as the objectively-measured walkability index or walkscore, and compare their associations to individually-reported PA levels.

## Discussion

The WALKI-MUC project addresses current challenges in walkability research [78] with CY by combining qualitative and quantitative research methods for subjective walkability measurement. This research contributes to walkability research in terms of both the methodology used and regarding the findings that are generated in the course of the project. The participatory method combination empowers CY to actively perceive their neighbourhood and at the same time yields valid information on urban environments. This knowledge can help to tailor scales like the NEWS-Y to local characteristics. Further, the engagement of CY and their parents in research or community projects may ultimately increase their sense of belonging. Due to the intuitiveness of photovoice activities, the methodology is universally applicable in communities all over the world including different settings with other target groups. In Germany, this study significantly advances environmental research with CY on the topic of PA. In addition to existing German indices that objectively measure PA-friendliness or walkability of urban areas, the NEWS-Y-G questionnaire offers the opportunity to include CY's perceptions through the assessment of subjective walkability. The results from Munich and potentially further German cities can be used for comparative analyses of different environments. Subjective walkability measurements may enrich objective methods, e.g. GIS, which are often used in urban planning because of their availability and economic advantage. Urban planners are provided with CY's perspective on how they perceive their environment regarding places that encourage them to exercise, play and be physically active. Implications about concrete actions for supporting PA in urban environments can be derived to promote this crucial health behaviour and help to prevent the young generation from potential detrimental effects of the progressing urbanisation.

## Acknowledgments

We thank Dr. Christoph Mall, who conceived the WALKI-MUC project in its first draft, as well as Prof. Dr. Andreas Humpe, Prof. Dr. Filip Mess and Katharina Sterr, who supported the project ever since with their advice. Further, we thank Georg Sedlmeir and the department for climate protection and environment of the city of Munich (RKU-UVO11) for providing the walkability index data.

## Author Contributions

**Conceptualization:** Daniel Alexander Scheller, Joachim Bachner.

**Methodology:** Daniel Alexander Scheller, Joachim Bachner.

**Project administration:** Daniel Alexander Scheller, Joachim Bachner.

**Resources:** Daniel Alexander Scheller, Joachim Bachner.

**Supervision:** Joachim Bachner.

**Validation:** Daniel Alexander Scheller, Joachim Bachner.

**Visualization:** Daniel Alexander Scheller.

**Writing – original draft:** Daniel Alexander Scheller.

**Writing – review & editing:** Joachim Bachner.

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
