## [Decision Letter · Decision Letter 0]

26 Jun 2023

PONE-D-23-04366

Subjective walkability perceived by children and adolescents living in urban environments: A study protocol for participatory methods and scale development in the WALKI-MUC project

PLOS ONE

Dear Authors,

Thank you for submitting your manuscript to PLOS ONE. After careful consideration, we feel that it has merit but does not fully meet PLOS ONE’s publication criteria as it currently stands. Therefore, we invite you to submit a revised version of the manuscript that addresses the points raised during the review process.

I really think that this paper is suitable for publication, but the authors need to modify according to the  reviewer.

We look forward to receiving your revised manuscript.

Kind regards,

Guglielmo Campus, Ph.D DDS

Academic Editor

PLOS ONE

3. We note that [Figure 1] in your submission contain [map/satellite] images which may be copyrighted. All PLOS content is published under the Creative Commons Attribution License (CC BY 4.0), which means that the manuscript, images, and Supporting Information files will be freely available online, and any third party is permitted to access, download, copy, distribute, and use these materials in any way, even commercially, with proper attribution. For these reasons, we cannot publish previously copyrighted maps or satellite images created using proprietary data, such as Google software (Google Maps, Street View, and Earth). For more information, see our copyright guidelines: http://journals.plos.org/plosone/s/licenses-and-copyright.

Natural Earth (public domain): http://www.naturalearthdata.com/.

Reviewers' comments:

Reviewer's Responses to Questions

**Comments to the Author**

1. Does the manuscript provide a valid rationale for the proposed study, with clearly identified and justified research questions?

Reviewer #1: Partly

Reviewer #2: Yes

2. Is the protocol technically sound and planned in a manner that will lead to a meaningful outcome and allow testing the stated hypotheses?

Reviewer #1: Yes

Reviewer #2: Yes

3. Is the methodology feasible and described in sufficient detail to allow the work to be replicable?

Reviewer #1: Yes

Reviewer #2: Yes

4. Have the authors described where all data underlying the findings will be made available when the study is complete?

Reviewer #1: Yes

Reviewer #2: Yes

5. Is the manuscript presented in an intelligible fashion and written in standard English?

Reviewer #1: Yes

Reviewer #2: Yes

6. Review Comments to the Author

You may also provide optional suggestions and comments to authors that they might find helpful in planning their study.

Reviewer #1: Thank you for the opportunity to review the study protocol entitled “Subjective walkability perceived by children and adolescents living in urban environments: A study protocol for participatory methods and scale development in the WALKI-MUC project”. The paper is well written, with an interesting research design that I think is suitable for publication. I think that, before publication, there are some issues that the authors may consider. My main concern is that, despite the authors have written a long introduction, there are some misleading concepts (on walking, physical activity, walkability…). Specifically, my suggestions are:

1. In the introduction, there is a need to better conceptualize the physical activity terminology. Physical activity is not the same as walking or exercise. This is particularly confusing in the headline “The young, their physical activity level and perceived environment” (L56). My suggestion is to start this section with a brief definition of walking and physical activity to justify why the study will focus on the walking environment (instead of a general PA-supportive environment). And if it is the case that the authors include broader aspects of physical activity beyond walking, there is a need to reframe the text (starting with the title).

2. In the same section (the one that starts in L56), the authors made a great effort to summarize the previous literature linking built environment features with YC physical activity, and also why subjective measures must be taken into account. However, despite all this information, there is no clear framework on how subjectiveness connects with the socio-ecological framework. My suggestion will be to include a figure that serves as a conceptual framework, adapting the socio-ecologic framework to the conceptual framework of this study (e.g. how do person characteristics modify the association between built environment features and YC physical activity? How perceptions are important in this?

3. I think there is some confusion with the concept of walkability. Despite some authors use walkability as a broader concept that can be applied to all supportive physical activity-friendly environments, there is evidence that the role of different environments impacts differently on physical activity, and thus, on health. For example, places, where YC might do exercise or vigorous physical activity, are not conceptually the same as places that favor active walking to school. This needs to be clarified in the introduction.

4. I am surprised that neither the introduction, objectives nor study design takes into account social inequalities, as they affect perceptions, where people live, and their physical activity. The only mention comes from the associations in objective 3. For example, why not consider neighborhood SES in the study design and area selection? How will the authors be sure that low-SES or ethnic minorities participate in the study? This is crucial, as this will create measures that will be biased against the privileged population.

5. L33: Regarding phase 1 I have a couple of questions that I think will not be difficult for the authors to clarify. (1) Are boys are girls in the same groups? Previous evidence suggests that (in adults) men can silence women’s discourses, and thus, many phtovoice projects separate them. (2) Why only an opening and a closing session? Previous research using photovoice usually have 4, 5 or 6 sessions to create collective discourses through different interactions.

6. L585. How will the authors deal with the potential same source bias regarding the association between neighborhood perceptions and self-reported physical activity?

Reviewer #2: The article describes in detail a protocol for a study of children and adolescents' perception of walkability. The work is very well organised and the methodological procedures used are well justified. The authors identify the contributions of the project results to the area of knowledge and to inform public policy on urban design. The combined use of qualitative research strategies is an innovative aspect of the protocol. The careful description will allow other studies to replicate the qualitative strategies in other contexts.

I would only suggest a paragraph in the introduction and abstract stating that this paper intends to describe the protocol of a study. Although it is in the title of the article, the introduction and abstract should also contain this information.

7. PLOS authors have the option to publish the peer review history of their article (what does this mean?). If published, this will include your full peer review and any attached files.

Reviewer #1: No

Reviewer #2: No

---

## [Author Response · Author response to Decision Letter 0]

16 Oct 2023

Reviewer #1 comments:

1. Authors’ response: We have considered your valuable feedback and restructured the introduction section accordingly. We now begin with a concise definition of physical activity and the relation to walking, which serves to justify our emphasis on studying the built environment in terms of a PA-promoting environment that can be reached by walking (L60-71). We emphasise that walking as a form of active recreation is beneficial in itself, but is also a healthy means to reach public places, exercise or leisure destinations. The following references have been added to the list:

Morris JN, Hardman AE. Walking to health. Sports Med 1997; 23(5):306–32.

Caspersen CJ, Powell KE, Christenson GM. Physical activity, exercise, and physical fitness: definitions and distinctions for health-related research. Public Health Rep 1985; 100(2):126–31

2. Authors’ response: Your thoughts touch on a crucial aspect of our study, thank you for this comment. We would like to refer to a recent study by Koohsari et al. (2023), which has analysed exactly these questions and demonstrated that place attachment dimensions were significantly associated to neighbourhood-specific physical activity in adults, particularly regarding walking for transport and recreation. Notably, the associations between place attachment and walking for transport in this study were mediated by individuals' perceptions of the built environment measured with the NEWS-A (L104-115). We believe that understanding the role of perceptions of/and attachment to PA-friendly places in the first study phase has significant implications for developing the NEWS-Y-G and measuring subjective walkability with this tool in the second study phase. In response to your insightful questions, we have created two graphics 1) a visualisation of the WALKI-MUC project (L331) and 2) a modification of the socio-ecological framework to illustrate the conceptual framework of our study (L341). A paragraph has been added to describe how the socio-ecological model and the framework of our study are interconnected (L335-340). The following reference has been added to the list:

Koohsari MJ, Yasunaga A, Oka K, Nakaya T, Nagai Y, McCormack GR. Place attachment and walking behaviour: Mediation by perceived neighbourhood walkability. Landscape and Urban Planning 2023; 235:104767.

3. Authors’ response: We genuinely appreciate your thoughtful feedback. Your comments have prompted us to introduce the concept of “walkability” as defined by Bucksch and Schneider (2014) in a broad and narrow sense. In the broad sense, walkability can encompass various physical activity-friendly environments including places that favor walking and places for exercise and play that can be reached by walking (L117-128). We now emphasize that spaces supporting PA, irrespective of the specific type they promote, are integral to a PA-friendly environment by describing the "third places framework" in more detail (L260-265). The following reference has been added to the list:

Bucksch J, Schneider S. Walkability – Einführung und Überblick. In: Bucksch J, Schneider S, editors. Walkability. Das Handbuch zur Bewegungsförderung in der Kommune.; 2014. p. 15–26.

4. Authors’ response: Thank you for this important comment. We have expanded our manuscript to include a paragraph that addresses the crucial aspect of social inequalities in PA environmental research (L273-277). In terms of participant recruitment, we are aware of this aspect and guarantee a diverse range of participants from different neighbourhoods and sociodemographic background by considering the data of the social monitoring system of the city government in the selection of districts that form part of the study (L358-363). In study phase one, we first contacted the education centers “Bildungslokale” that are networked with all types of schools and represent an essential contact point for families with a migration background (L372-375). The following references have been added to the list:

Page AS, Cooper AR, Griew P, Jago R. Independent mobility, perceptions of the built environment and children's participation in play, active travel and structured exercise and sport: the PEACH Project. Int J Behav Nutr Phys Act 2010; 7:17.

Brockman R, Jago R, Fox KR, Thompson JL, Cartwright K, Page AS. "Get off the sofa and go and play": family and socioeconomic influences on the physical activity of 10-11 year old children. BMC Public Health 2009; 9:253.

Landeshauptstadt München S. Monitoring für das Sozialreferat: Tabellenband 2021 2022. Available from: URL: https://stadt.muenchen.de/dam/jcr:db6c3f41-2f01-4341-9538-e91072b1886c/SOZ_Monitoring-Muenchen-2021.pdf.

5. Authors’ response: We appreciate your thoughtful questions regarding phase 1 of our study and we are pleased to provide further insights into our approach (L389-390) and how it may be adjusted after the pilot study (L528-523). Our decision to conduct only one opening and one closing session is largely influenced by practical considerations and by the expected outcomes of the participatory method combination (L398-405).

6. Authors’ response: Thank you for the advice to consider the same-source bias in our analysis of associations between subjective walkability measured with the NEWS-Y-G and physical activity. A detailed description of three strategies to deal with this bias based on previous literature can now be found in the methods section (L648-656). The following references have been added to the list:

Chum A, O'Campo P, Lachaud J, Fink N, Kirst M, Nisenbaum R. Evaluating same-source bias in the association between neighbourhood characteristics and depression in a community sample from Toronto, Canada. Soc Psychiatry Psychiatr Epidemiol 2019; 54(10):1177–87.

Mujahid MS, Diez Roux AV, Morenoff JD, Raghunathan T. Assessing the measurement properties of neighborhood scales: from psychometrics to ecometrics. Am J Epidemiol 2007; 165(8):858–67

Reviewer #2 comments:

1. Authors’ response: We greatly appreciate your positive feedback on our article. We are especially glad that you highlighted the innovative aspect of our study, which combines various qualitative research strategies. Our aim is to not only generate valuable insights but also to provide a replicable framework that can be adapted and utilised in various contexts. All changes are tracked and the line references in the comment refer to the text with a simple mark-up.

Editor comments:

1. Authors’ response: We ensured that our manuscript meets PLOS ONE's style requirements, including those for file naming.

2. Authors’ response: Thank you for the note about the disparity in the grant information provided in the ‘Funding Information’ and ‘Financial Disclosure’ sections. The Bavarian Ministry of Health and Care is responsible for the public health service, for hygiene and infection control as well as for medicine and pharmacy. It is supported in these matters by the public health offices and the Bavarian State Office for Health and Food Safety. We receive our funding directly from the latter, the Bavarian State Office for Health and Food Safety, and only indirectly from the Bavarian Ministry of Health and Care. The financial disclosure statement has to be changed accordingly as follows: 

“The WALKI-MUC project is funded by the Bavarian State Office for Health and Food Safety (Bayerisches Landesamt für Gesundheit und Lebensmittelsicherheit; https://www.lgl.bayern.de/). The funders did not and will not have a role in study design, data collection and analysis, decision to publish, or preparation of the manuscript.”

3. Authors’ response: Thank you for indicating resources for replacing our map with a copyrighted map. Figure 1 was created based on OpenStreetMap®. OpenStreetMap® is open data, licensed under the Open Data Commons Open Database License from the OpenStreetMap Foundation (OSMF). Their documentation is licensed under the Creative Commons Attribution-ShareAlike 2.0 License (CC BY-SA 2.0). Consequently, we decided to leave Figure 1 (map of Munich) out of the manuscript.

4. Authors’ response: We reviewed and updated our reference list. There has been one change in the reference list due to a new demographic report for Munich that has been published in the meantime. We have adjusted the percentage for the projected population growth until 2040 accordingly from 16% to 14% (L281-282) and added the new report to the reference list. All other changes to the reference list refer to the reviewer's comments.

Landeshauptstadt München. Demografiebericht München – Teil 1 Analyse 2022 und Bevölkerungsprognose 2023 bis 2040 für die Landeshauptstadt; 2023. Available from: URL: https://stadt.muenchen.de/dam/jcr:934018c0-0a9e-47a2-aea6-0dafcf829ea8/LHM_Demografiebericht-Teil1_2023.pdf.

---

## [Decision Letter · Decision Letter 1]

7 Feb 2024

Subjective walkability perceived by children and adolescents living in urban environments: A study protocol for participatory methods and scale development in the WALKI-MUC project

PONE-D-23-04366R1

Dear Mag. Scheller,

We’re pleased to inform you that your manuscript has been judged scientifically suitable for publication and will be formally accepted for publication once it meets all outstanding technical requirements.

Kind regards,

Michał Suchanek, D.Sc.

Academic Editor

PLOS ONE

Additional Editor Comments (optional):

Given the current quality of the article and the overall process since submission, I suggest accepting the article.

Reviewers' comments:

Reviewer's Responses to Questions

**Comments to the Author**

1. Does the manuscript provide a valid rationale for the proposed study, with clearly identified and justified research questions?

Reviewer #1: Yes

2. Is the protocol technically sound and planned in a manner that will lead to a meaningful outcome and allow testing the stated hypotheses?

Reviewer #1: Yes

3. Is the methodology feasible and described in sufficient detail to allow the work to be replicable?

Reviewer #1: Yes

4. Have the authors described where all data underlying the findings will be made available when the study is complete?

Reviewer #1: Yes

5. Is the manuscript presented in an intelligible fashion and written in standard English?

Reviewer #1: Yes

6. Review Comments to the Author

You may also provide optional suggestions and comments to authors that they might find helpful in planning their study.

Reviewer #1: Thank you very much for the opportunity to review the revised version of the manuscript “Subjective walkability perceived by children and adolescents living in urban environments: A study protocol for participatory methods and scale development in the WALKI-MUC project”. I would like to thank the authors for their response. They have taken their time to answer every question that I had on the original submission. I find this version with a stronger conceptual background, and I am waiting to see the results of this study.

7. PLOS authors have the option to publish the peer review history of their article (what does this mean?). If published, this will include your full peer review and any attached files.

Reviewer #1: No

---

## [Editor Report · Acceptance letter]

23 Feb 2024

PONE-D-23-04366R1 

PLOS ONE

Dear Dr. Scheller, 

I'm pleased to inform you that your manuscript has been deemed suitable for publication in PLOS ONE. Congratulations! Your manuscript is now being handed over to our production team.

Kind regards, 

on behalf of

Dr. Michał Suchanek 

Academic Editor

PLOS ONE